# Sequence specificity analysis of the SETD2 protein lysine methyltransferase and discovery of a SETD2 super-substrate

Maren Kirstin Schuhmacher[1,4], Serap Beldar[2,4], Mina S. Khella[1,3,4], Alexander Bröhm[1], Jan Ludwig[1], Wolfram Tempel[2], Sara Weirich[1], Jinrong Min[2✉] & Albert Jeltsch [1✉]

SETD2 catalyzes methylation at lysine 36 of histone H3 and it has many disease connections. We investigated the substrate sequence specificity of SETD2 and identified nine additional peptide and one protein (FBN1) substrates. Our data showed that SETD2 strongly prefers amino acids different from those in the H3K36 sequence at several positions of its specificity profile. Based on this, we designed an optimized super-substrate containing four amino acid exchanges and show by quantitative methylation assays with SETD2 that the super-substrate peptide is methylated about 290-fold more efficiently than the H3K36 peptide. Protein methylation studies confirmed very strong SETD2 methylation of the super-substrate in vitro and in cells. We solved the structure of SETD2 with bound super-substrate peptide containing a target lysine to methionine mutation, which revealed better interactions involving three of the substituted residues. Our data illustrate that substrate sequence design can strongly increase the activity of protein lysine methyltransferases.

[1] Institute of Biochemistry and Technical Biochemistry, University of Stuttgart, Allmandring 31, 70569 Stuttgart, Germany. [2] Structural Genomics Consortium, University of Toronto, 101 College Street, Toronto, ON M5G 1L7, Canada. [3] Biochemistry Department, Faculty of Pharmacy, Ain Shams University, African Union Organization Street, Abbassia, Cairo 11566, Egypt. [4] These authors contributed equally: Maren Kirstin Schuhmacher, Serap Beldar, Mina S. Khella. ✉email: jr.min@utoronto.ca; albert.jeltsch@ibtb.uni-stuttgart.de

Numerous covalent post-translational modifications (PTMs) occur on the tails of histone proteins including acetylation of lysine residues, methylation of lysine and arginine residues, and phosphorylation of serine or threonine residues[1]. Together with other chromatin modifications, histone PTMs play very important roles in the regulation of chromatin structure and gene expression[2–4]. The methylation of the ε-amino group of lysine residues is catalyzed by protein lysine methyltransferases (PKMTs), which can introduce up to three methyl groups with high specificity using S-Adenosyl-L-methionine (AdoMet) as methyl group donor. One group of PKMTs contains a SET (Su(var) 3–9, Enhancer of Zeste (E(z)) and Trithorax (trx)) domain, which harbors the catalytically active center for the transfer of the methyl group[5–7].

The SET-domain containing protein 2 (SETD2, also called KMT3A) PKMT was first identified as huntingtin interacting protein 1 (HYPB, HIP-1)[8]. It has a size of 230 kDa corresponding to 2564 amino acids and can add up to three methyl groups to K36 of histone H3[9]. The ability for trimethylation of H3K36 makes SETD2 unique, because no other trimethyltransferase for this lysine residue is known in human cells[9,10]. The structures of SETD2 in complex with H3K36 variant peptides and S-Adenosyl-L-homocysteine (AdoHcy) revealed that the substrate peptide is positioned in a deep binding channel in the catalytic SET domain where several contacts between the enzyme and the substrate peptide are formed[11,12]. H3K36me3 is enriched in the gene bodies of expressed genes, because SETD2 is recruited to its genomic target sites by the elongating RNA Polymerase II[3,8,13,14]. It has a role in the maintenance of repressed chromatin in transcribed regions by recruitment of histone deacetylases, the H3K4 demethylase LSD2, chromatin remodelers, and DNA methylation via the PWWP domain of DNMT3 enzymes. Moreover, H3K36me3 in gene bodies has been associated with alternative splicing[3,13,14].

Many studies have shown that lysine methylation not only occurs on histones, but it is a very abundant PTM on non-histone proteins as well, where it has diverse regulatory roles[15–17]. Park et al.[16] showed that SETD2 methylates α-tubulin at lysine 40 and that SETD2 is crucial for mitosis and cytokinesis[18]. Later studies reported that the signal transducer and activator of transcription 1-alpha/beta (STAT1) is also methylated by SETD2 at lysine 525 and this modification is an important signal for IFNα-dependent antiviral immune response[19].

Previous investigations revealed many somatic mutations of the SETD2 gene in cancer tissues, for example, in pediatric and adult high grade gliomas[20]. Another cancer type in which SETD2 plays an important role is Sporadic clear cell renal cell carcinoma (cRCC), where frameshift, non-sense and missense mutations in SETD2 were observed, suggesting a loss-of-function mechanism[21]. This model was supported by the observation that the level of H3K36 trimethylation is reduced in cRCC, but H3K36me2 is not affected[22]. Another mechanism of SETD2 inhibition in cancer cells is the expression of the so-called H3K36M oncohistone, frequently found in chondroblastomas[23]. Structural and kinetic studies have shown that the H3K36M mutation inhibits the methyltransferase activity of SETD2, because the methionine residue binds into the lysine binding channel of the active center and thereby functions as a competitive enzyme inhibitor[11]. The inhibition of H3K36 PKMTs by the K36M oncohistone leads to massive perturbations of genome wide H3K36me3 and H3K27me3 patterns[11,24]. Genetic mutations of SETD2 cause an overgrowth syndrome[25] related to the SOTOS syndrome, which is majorly caused by mutations in the H3K36 mono- and dimethyltransferase NSD1[26].

While the crystal structures of SETD2 in complex with H3 peptides provide the interaction details between the SETD2 substrate binding channel and the bound peptide[11,12], the substrate specificity of SETD2 for peptide methylation has not yet been systematically analyzed. Using a peptide array based approach, we have determined the amino acid specificity profile of the substrate sequence for SETD2 in this study. Based on this information, we identified 9 novel and strongly methylated non-histone peptide substrates and a methylation site at K666 of Fibrillin-1 at the protein level. Strikingly, we observed that SETD2 prefers other amino acids than those present in the H3K36 sequence at several places of the specificity profile. Based on this observation, a super-substrate of SETD2 was designed and shown to be methylated much more efficiently than H3K36 in vitro and in cells. Structural analysis pinpointed improved molecular interactions as potential reasons for the increased methylation efficiency of the super-substrate. However, the human proteome does not contain a protein fully matching the optimized SETD2 substrate sequence indicating that additional work will be needed to understand this unexpected property of SETD2.

## Results

**SETD2 expression and purification**. The His$_6$-tagged catalytic SET domain of SETD2 was expressed in E. coli cells and purified through Ni-NTA-beads (Supplementary Fig. 1a). To examine the activity of SETD2, the enzyme was incubated with reconstituted mononucleosomes (rec. MN), native mononucleosomes isolated from HEK293 cells (nat. MN) and recombinant H3.1 protein as substrates in methylation buffer supplemented with radioactively labeled [methyl-$^3$H]-S-Adenosyl-L-methionine (AdoMet) as cofactor. After the methylation reaction, the samples were separated by SDS-PAGE and the transfer of the radioactively labeled methyl groups from the cofactor to the respective target lysine was visualized by autoradiography (Supplementary Fig. 1b). While no activity could be detected on the recombinant H3.1 protein, strong methylation was observed with the nucleosomal substrates. The activity of SETD2 on reconstituted mononucleosomes was higher than on the native mononucleosomes, presumably because the native mononucleosomes were already partially methylated at K36.

Next, the SETD2 activity was tested on peptide SPOT arrays containing fifteen amino acid long peptides immobilized on a cellulose membrane. As reported SETD2 substrates, the H3K36 (29–43), α-tubulin (33–47) and STAT1 (518–532) peptides (with the target lysine centered in each case) were included, and peptides with the corresponding target lysine mutated to alanine were used as negative controls. The peptide array was incubated in methylation buffer supplemented with SETD2 and radioactively labeled AdoMet. The methylation of the peptides was visualized by autoradiography (Supplementary Fig. 1c) showing detectable methylation of the H3K36 (29–43) peptide spot. The K36A spot and the other investigated peptide substrates (α-tubulin and STAT1) did not show a detectable peptide methylation signal. These data confirm the methylation activity of SETD2 at the H3K36 site in nucleosomal and peptide substrates. Methylation of the previously reported non-histone substrates α-tubulin and STAT1 was not detectable at the peptide level and, therefore, they were excluded from further investigation.

**Substrate specificity analysis of SETD2 in an H3K36 context**. After confirmation of the SETD2 activity, its substrate specificity was determined by using SPOT peptide arrays[27]. The H3K36 (aa 29–43) fragment was used as a template sequence in this analysis, and each single position of the template sequence was exchanged against 18 other natural amino acids creating single amino acid mutants for each position. Trp was excluded in this analysis

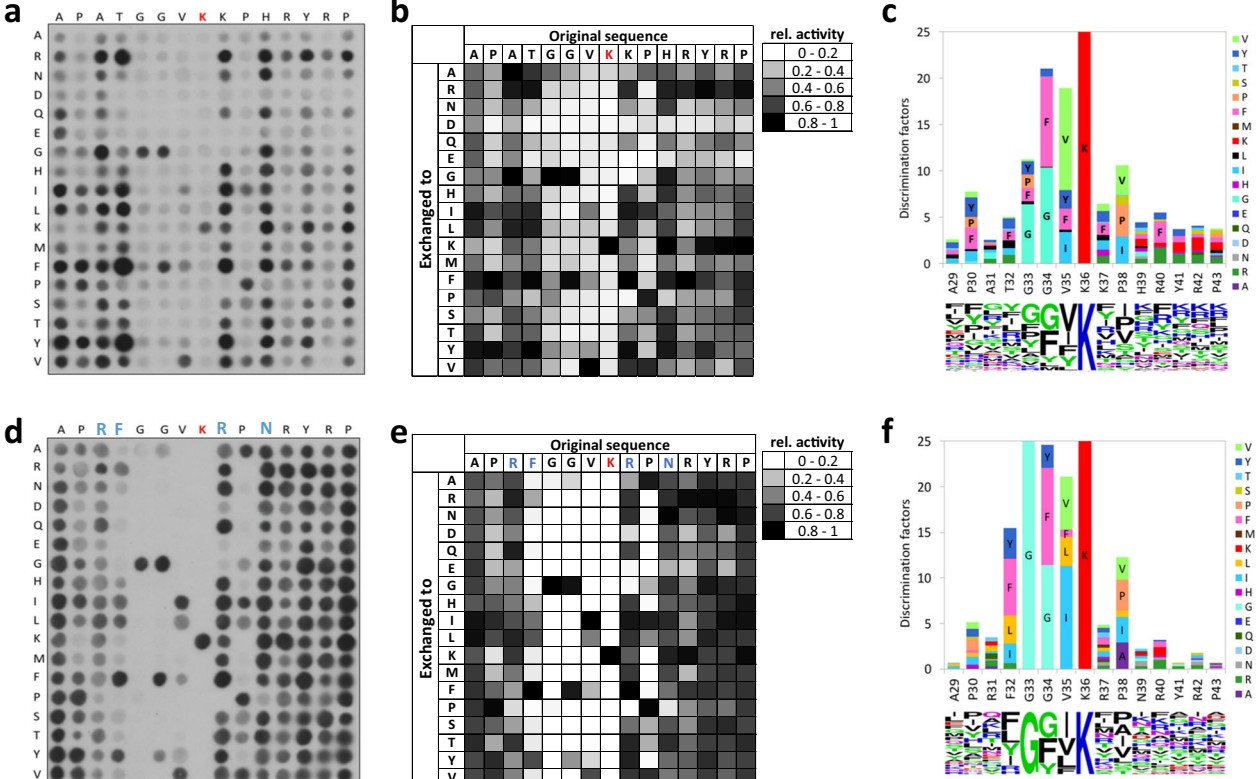

**Fig. 1 Peptide SPOT array substrate sequence specificity investigation of SETD2. a** Substrate sequence specificity scan using H3K36 (29–43) as template sequence. The template sequence is represented on the horizontal axis, K36 is printed in red. The vertical axis represents the 18 amino acids, which were used to create single amino acid mutants of the template sequence. The peptide array was methylated with SETD2 using radioactively labeled AdoMet and the transfer of methyl groups was detected by autoradiography. **b** Analysis of two independent specificity scan experiments. The signals of both independent replicates were quantified, normalized and averaged. **c** Discrimination factors showing the preference of SETD2 for specific amino acids over all others at each position[27] and visualization of the substrate specificity motif of SETD2 as Weblogo (https://weblogo.berkeley.edu/logo.cgi)[29]. **d**–**f** Substrate sequence specificity investigation of SETD2 based on the super-substrate as template sequence. Residues changed in the super-substrate are indicated in blue. For errors of individual spots and error distributions refer to Supplementary Fig. 2.

because of its unfavorable coupling properties. Cys was omitted because it can be methylated itself[28]. This setup resulted in 270 peptide spots whose methylation could be investigated in a single experiment (Fig. 1a). The methylation experiments of the peptide arrays were performed as described above in duplicates, which were normalized and averaged (Fig. 1b). The standard deviations (SD) of the methylation activity of each single spot were calculated, which showed that most spots had an SD smaller than ±10% (Supplementary Fig. 2a) indicating a high reproducibility of the results. For visualization of the substrate specificity motif of SETD2, discrimination factors were calculated as previously described[27]. The discrimination factors show the preference of SETD2 for one specific amino acid over all other amino acids in each single position of the peptide sequence (Fig. 1c). In addition, a Weblogo of the substrate specificity profile was calculated as well[29].

Our data revealed that SETD2 is a highly specific PKMT with specific sequence readout between P30 (position −6, if K36 is annotated as position 0) and R42 (+6). The strongest specificity was observed at G33 (−3), G34 (−2), V35 (−1) and P38 (+2). Surprisingly, we observed that SETD2 prefers amino acids different from the canonical sequence surrounding K36 in H3 at five sites of the profile. At position −5, SETD2 prefers R, G, and L over the naturally occurring alanine residue in the H3K36 substrate. At the −4 position, F, Y, L, I, and R are more preferred than the natural T. At position +1, SETD2 has a higher preference for R, H, I, L, F, Y, and V compared to the natural K.

At position +3, R, N, G, K, F, and Y are more favored than the naturally occurring H, and finally R and K are preferred over Y at position +5.

**Design of a super-substrate for SETD2.** Based on these observations, we aimed to explore, if better substrates than H3K36 can be designed for SETD2. We synthesized peptide SPOT arrays including H3K36 (29–43) as a positive control and the H3K36A mutant as a negative control, in which up to 5 combined mutations were systematically introduced that converted the original H3K36 residues into residues that are more favored (Fig. 2a and Supplementary Table 1). This peptide array was methylated by SETD2 as described above and the methylation signal was captured by autoradiography. Strikingly, we observed that most of the mutated peptides were indeed more strongly methylated than the H3K36 peptide indicating that the mutant peptides are preferred substrates. We observed a stepwise improvement of activity with increasing number of mutations starting with T32F, going over A31R/T32F, A31R/T32F/K37R, and finally to A31R/T32F/K37R combined with H39N/K. Among the investigated peptides, the peptide sequence of spot B10 was selected as a super-substrate (ssK36) peptide, because it was one of the most strongly methylated peptides (sequence: A P R F G G V K R P N R Y R P, with the altered amino acids printed in blue; the red labeled residue represents the target lysine K36). The Y to R exchange at the +5 site was not included to avoid the generation of a triple-R

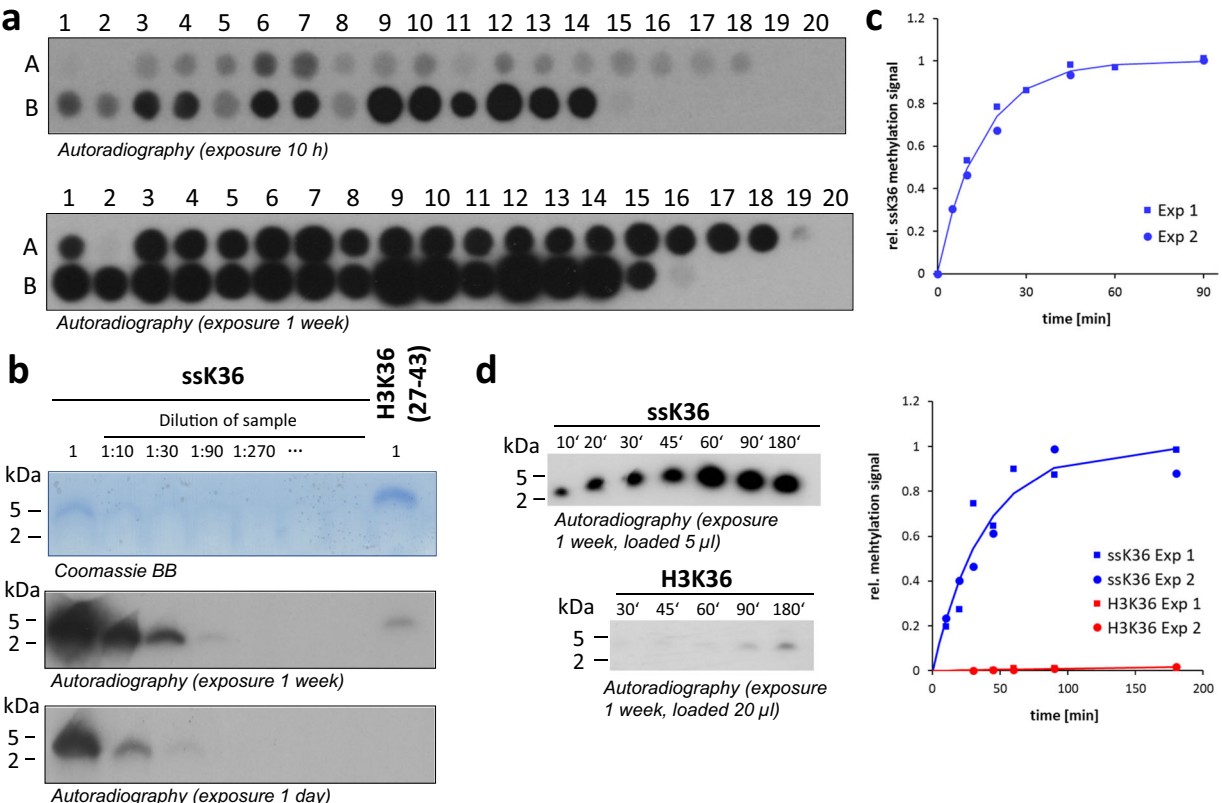

**Fig. 2 Design of an optimized SETD2 peptide substrate and analysis of its methylation. a** Variants of the H3K36 (29–43) peptide containing up to five mutations (Supplementary Table 1) were synthesized on a cellulose membrane and methylated by SETD2. The signal of the transferred radioactive methyl groups was detected by autoradiography. Spots A1 and B15 contain the H3K36 sequence, A2 and B16 the H3K36A peptide. Two films with different exposure times are shown to visualize weaker methylated spots as well. **b** Comparison of the methylation of purified super-substrate (ssK36) and the H3K36 peptides. The upper panel shows a Tricine-SDS-gel of the ssK36 and H3K36 peptides stained with Coomassie BB. Methylation reactions were conducted using 20 μM peptide and 6 μM enzyme. The lower panels display autoradiography films after different exposure times allowing to compare the methylation levels of the different samples. For comparison, different dilutions of the methylated ssK36 were loaded as indicated. **c** Time course of ssK36 peptide methylation determined under single turnover conditions using 5 μM substrate and 6 μM enzyme. Fit of the data to an exponential reaction progress curve revealed a single turnover rate constant of 4.07 ± 0.39 h$^{-1}$ (average ± SEM). **d** Direct comparison of the time courses of methylation of the ssK36 and H3K36 peptides determined using 20 μM substrate and 6 μM enzyme. The gels with the radioactive ssK36 (5 μl loaded per well) and H3K36 peptides (20 μl loaded per well) were exposed together to the same x-ray film. The figure shows exemplary autoradiography images and the quantitative analysis of two experiments revealing a 290 ± 70 (average ±SEM) higher initial slope for ssK36 methylation. An enlargement of the H3K36 data with adjusted y-axis scale is shown in Supplementary Fig. 3d.

sequence and because it did not result in a clear increase in activity. Densitometric analysis of the intensities and shapes of 5 pairs of H3K36 and ssK36 peptide spots on independent arrays revealed a 70 ± 10 (SD) fold stronger methylation of ssK36 than H3K36.

**Peptide methylation and inhibition studies with ssK36**. To compare the methylation rates of the H3K36 and super-substrate peptides directly, purified ssK36 and H3K36 were methylated by SETD2, resolved on a Tricine gel and the methylation activity captured by autoradiography (Fig. 2b). To allow direct comparison of the methylation levels of both substrates, methylated ssK36 was loaded in different dilutions on the same gel. Densitometric analysis of three independent experiments showed that the ssK36 peptide was methylated 50 ± 4 (SD) fold more efficiently than the wild type H3K36 peptide, which is in good qualitative agreement with the estimations from the peptide spot methylation experiments.

To further characterize the methylation kinetics of the super-substrate peptide, steady-state methylation experiments were

performed using the super-substrate peptide in different concentrations ranging from 20 to 320 μM (Supplementary Fig. 3a). The derived data were fitted to the Michaelis-Menten model revealing a $K_M$ of 45 μM. Hence, the apparent $K_M$ value of the methylation of the super-substrate was only slightly improved when compared with the $K_M$ of the H3K36 peptide previously determined under the same conditions (65 μM[30]) suggesting that most of the methylation rate enhancement observed with the ssK36 is due an increase in $k_{cat}$. It has been previously shown that H3K36M is a substrate competitive inhibitor of SETD2[11,30]. To further study the difference in peptide binding, inhibition of SETD2 by ssK36M and H3K36M inhibitor peptides was investigated (Supplementary Fig. 3b and c). The results were fitted to a competitive inhibition model revealing $K_i$-values of 128 μM for the ssK36M peptide and 447 μM for H3K36M indicating a roughly 3.5-fold increase in binding of the ssK36M peptide.

To determine the methylation rates of H3K36 and ssK36 more quantitatively, the time course of ssK36 methylation was determined under single turnover conditions revealing a methylation rate constant of $k_{st}$ = 4.07 h$^{-1}$ (Fig. 2c). Finally, the

time courses of methylation of ssK36 and H3K36 were determined under matching conditions and quantified together by joined exposition of the gels to the same x-ray film (Fig. 2d and Supplementary Fig. 3d). Comparison of the initial slopes of the methylation reactions revealed that ssK36 was methylated 290-fold faster than H3K36.

**Substrate specificity analysis of SETD2 in an ssK36 context.** Based on the sequence of the super-substrate peptide, a second substrate specificity analysis was performed (Fig. 1d–f and Supplementary Fig. 2b). In this experiment, only few amino acid exchanges led to slightly increased methylation efficiencies (most prominently I instead of V at position −1 and R instead of Y at the +5 site). This finding confirms that the super-substrate sequence is close to the optimal recognition sequence of SETD2. As before, a very specific interaction with the target peptide was observed. At the −4 position, SETD2 only accepts F, R, I, L, and Y. At the −3 position, SETD2 is highly specific and only tolerates G. At position −2, a strong preference of SETD2 was observed as well, where four residues are allowed with F and G being favorably accepted and Y and A to a lesser extent. At position −1, I is stronger recognized by SETD2 than V. In addition, L, F, and Y are weakly tolerated at this position. At position +1, E, G, and P are not tolerated. The position +2 is also highly specifically recognized and only A, G, I, L, S, T, and V are tolerated apart from P, with clear preferences for I, A, and P. Altogether, the following specificity motif based on the super-substrate (29–43) sequence of SETD2 was determined (the amino acids of the ssK36 peptide are underlined and red represents the target lysine 36):

(R, I, L, Y, F) - $\underline{G}$ - (F, $\underline{G}$ > A, Y) - (I, $\underline{V}$, L, F, Y) - K - R (not E, G, P) - (A, I, $\underline{P}$ > G, L, S, T, V)

Interestingly, both substrate specificity profile analyses resulted in very similar amino acid preferences, except that the phenylalanine at the −4 position and glycine at the −3 position showed stronger readout in the more favorable super-substrate context than the corresponding residues in H3K36.

**Identification of SETD2 non-histone peptide substrates.** A Scansite database search with the super-substrate peptide sequence revealed that this sequence is not present in the human proteome. However, we could identify 166 possible targets, which contain partial matches to the substrate specificity profile. The possible methylation of these 166 putative substrates was investigated at the peptide level. For this, fifteen amino acid long peptides containing the sequence of the non-histone targets surrounding the predicted target lysine were synthesized on a membrane. The H3K36 (29–43) peptide (spots A1, I9), the super-substrate peptide (spots A3, I11) and the corresponding lysine to alanine mutation peptides (spot A2, I10, and A4, I12) were included as positive and negative controls (Fig. 3a, Supplementary Fig. 4 and Supplementary Table 2). The methylation reactions and the detection of the methylation signals were performed in duplicates as described above. We observed that all of the investigated peptides were more weakly methylated than the super-substrate peptide. However, several of the methylated non-histone substrates were methylated at least as strong as the H3K36 peptide suggesting that their methylation could be relevant in cells. In the next step, 20 of the peptides were selected for further investigation based on the methylation intensity and the known functions of the proteins to confirm the methylation at the predicted target site. The peptides were synthesized on another membrane, now including the target lysine to alanine mutation

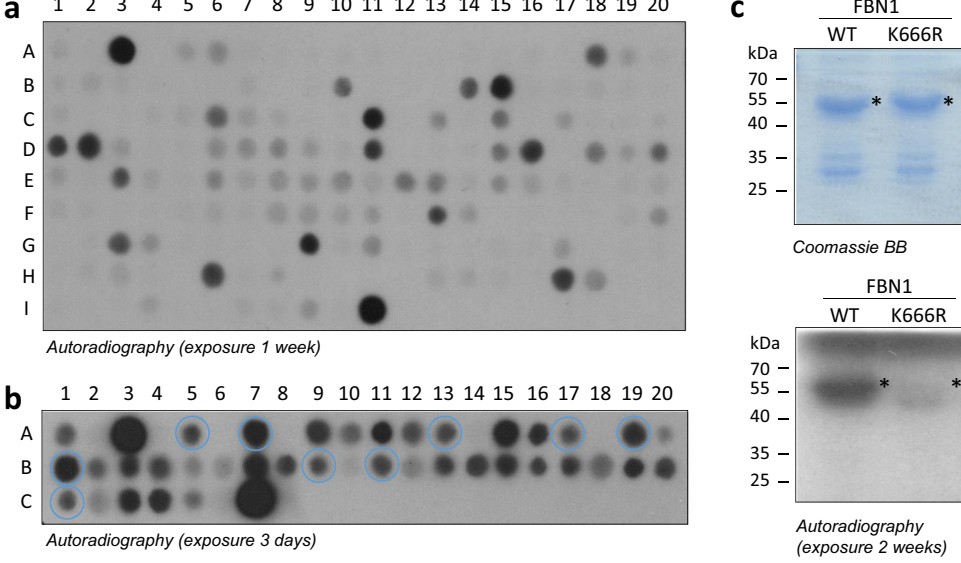

**Fig. 3 Identification of non-histone substrates of SETD2. a** Candidate non-histone peptide substrates were identified in the human proteome based on the SETD2 super-substrate specificity profile. The peptides were synthesized on a peptide SPOT array (Supplementary Table 2) and methylated by SETD2. As controls, the H3K36, H3K36A, ssK36 and ssK36A peptides were included on spots A1–A4 and I9–I12. The autoradiography image is an example of the obtained results. Note the strong methylation of several spots when comparing with the H3K36 control spots (A1 and I9). The quantification of the data of two experiments is shown in Supplementary Fig. 4. **b** Analysis of target lysine methylation of selected non-histone substrate peptides. Peptide SPOT arrays were synthesized containing the non-histone peptide substrates identified in panel A in addition to the lysine to alanine mutations presented on the array as every second spot (Supplementary Table 3). As controls, the H3K36, H3K36A, ssK36, and ssK36A mutant peptides were included on spot A1–A4 and C5–C8. The autoradiography image is an example of the obtained results. The quantification of the data of three experiments is shown in Supplementary Fig. 5. Peptides selected for furhter analysis are highlighted by blue circles. **c** Protein methylation of FBN1 and confirmation of the target lysine methylation. Both wild type (WT) and mutant (K666R) FBN1 were purified and equal amounts were methylated by SETD2 (2 μM). The upper panel shows a Coomassie BB stained SDS-gel the lower panel an autoradiography image. Loss of the methylation of the K666R mutant indicates that methylation occurred at the predicted target lysine K666. The size of FBN1 is indicated by an asterisk.

**Table 1 Non-histone protein substrates and H3K36-GST protein variants used for SETD2 methylation.**

| Protein name | Abbre-viation | Uni-prot no. | Target lysine | Peptide sequence | Spot no. in Fig. 3a | Spot no. in Fig. 3b | Cloned domain | MW (kDa) |
|---|---|---|---|---|---|---|---|---|
| H3K36 (29–43) | H3K36 | P68431 | 36 | APATGGV**K**KPHRYRP | A1, I9 | A1, C5 | 29–43 | 28 |
| ssK36 (29–43) | ssK36 | – | 36 | APRFGGV**K**RPNRYRP | A 3, I11 | A3, C7 | 29–43 | 28 |
| ssK36M (29–43) | ssK36M | – | – | APRFGGV**M**RPNRYRP | | | 29–43 | 28 |
| Ankyrin and armadillo repeat-containing protein | ANKAR | Q7Z5J8 | 317 | RRGIGYL**K**LICFLIP | A18 | A5 | n.a. | n.a. |
| Voltage-dependent T-type calcium channel subunit alpha-1G | CAC1G | O43497 | 804 | YGPFGYI**K**NPYNIFD | B10 | A7 | n.a. | n.a. |
| Collagen alpha-1 (XXII) chain | COMA | Q8NFW1 | 472 | SEQIGFL**K**TINCSCP | C6 | A13 | 429–603 | 46 |
| Dysferlin | DYSF | O75923 | 338 | AGARGYL**K**TSLCVLG | C15 | A17 | 218–516 | 58 |
| Fibrillin-1 | FBN1 | P35555 | 666 | STCYGGY**K**RGQCIKP | D1 | A19 | 529–763 | 52 |
| Fibrillin-2 | FBN2 | P35556 | 711 | STCYGGI**K**KGVCVRP | D2 | B1 | n.a. | n.a. |
| Hemicentin-1 | HMCN1 | Q96RW7 | 127 | EMSIGAI**K**IALEISL | D17 | B9 | 5–217 | 50 |
| Integrator complex subunit 6 | INT6 | Q9UL03 | 369 | GHPFGYL**K**ASTALNC | D20 | B11 | 296–463 | 45 |
| CMP-N-acetyl-neuraminate-poly-alpha-2,8-sialyltransferase | SIA8D | Q92187 | 350 | LHNRGAL**K**LTTGKCV | H6 | C1 | 98–358 | 55 |

The target lysine residues are printed in bold and underlined. MW refer to the molecular weight of the cloned GST fusion proteins.

control peptides in each case. The methylation reactions and the detection of the signal were conducted as described above (Fig. 3b, Supplementary Fig. 5 and Supplementary Table 3). This analysis confirmed strong methylation at the target site for 9 of the novel non-histone target peptides (Table 1).

**Investigation of non-histone protein methylation by SETD2.** The 9 new putative non-histone substrates of SETD2 were cloned into a pGEX-6p2 bacterial expression vector for protein expression. Six of them could be expressed and purified through glutathione sepharose beads (Table 1). For comparison, the super-substrate sequence was cloned with a C-terminal GST-tag (ssK36-GST) to mimic the natural setting of the H3K36 site in the N-terminal tail of histone 3. Similar amounts of the purified non-histone substrate proteins and the ssK36-GST protein were loaded on an SDS-gel and stained with Coomassie BB (Supplementary Fig. 6a). Afterward protein methylation reactions were conducted using radioactively labeled AdoMet and the radioactive signal of the methyl group transfer was detected by autoradiography (Supplementary Fig. 6b). Our data revealed methylation of the FBN1 protein, although more weakly than ssK36-GST. Next, we aimed to confirm the FBN1 methylation at the predicated target lysine K666. For this, a K666R mutation was introduced and the wildtype and mutant FBN1 proteins were purified by affinity chromatography. Equal amounts of wildtype and mutant FBN1 proteins were methylated by SETD2, revealing a clear methylation signal of FBN1 that was lost in the FBN1 K666R mutant (Fig. 3c). This result confirms SETD2 methylation of the FBN1 protein at K666.

**Crystal structure of the SETD2-ssK36M peptide complex.** To understand the molecular basis of the super-substrate methylation by SETD2, we determined the crystal structure of the human SETD2 catalytic domain (aa 1435–1711) bound to AdoHcy and the super-substrate peptide containing a target lysine to methionine mutation (ssK36M) (Table 2). The SETD2 catalytic domain comprises an N-terminal AWS motif coordinating two zinc ions (aa 1494–1548), as well as SET (aa 1550–1667) and post-SET (aa 1674–1690) subdomains[31]. The SET domain

**Table 2 Data collection and refinement statistics.**

SETD2 in complex with a H3-variant peptide (PDB 6VDB)

| | |
|---|---|
| **Data collection** | |
| Space group | P2₁2₁2₁ |
| Cell dimensions | |
| $a, b, c$ (Å) | 60.4,76.5,77.6 |
| $\alpha, \beta, \gamma$ (°) | 90, 90, 90 |
| Resolution (Å) | 47.65-2.30 (2.38-2.30) |
| $R_{merge}$ overall | 0.142 (1.216) |
| $R_{meas}$ overall | 0.153 (1.312) |
| Number Reflections | 16627 (1612) |
| I/σ(I) | 10.5 (1.7) |
| CC1/2 | 0.997 (0.664) |
| Completeness (%) | 100.0 (100.0) |
| Redundancy | 7.0 (7.1) |
| **Refinement** | |
| Resolution (Å) | 47.65-2.30 |
| Reflections/free | 16545/735 |
| Rwork/Rfree | 0.212/0.263 |
| All (no. atoms/mean B) | 2015/46.5 |
| Protein | 1840/46.4 |
| Peptide Ligand | 99/51.8 |
| Cofactor (AdoHcy) | 26/39.1 |
| Water | 24/39.7 |
| Others | 26/41.1 |
| R.m.s. deviations | |
| Bond lengths (Å) | 0.010 |
| Bond angles (°) | 1.1 |

The values in parentheses are for the highest-resolution shell.

assumes a triangular barrel-like structure (β1–β2; β3–β8–β7; β4–β6–β5–α6) as previously reported[11,12,31] (Fig. 4a). We observed clear electron density for the ssK36M peptide and a cofactor product corresponding to AdoHcy, feasibly formed from AdoMet under the crystallization conditions. The ssK36M peptide is in an extended conformation, and AdoHcy is surrounded by residues of the SET and post-SET domains (Fig. 4a, b, Supplementary Fig. 7). The overall SETD2 protein structure as well as the

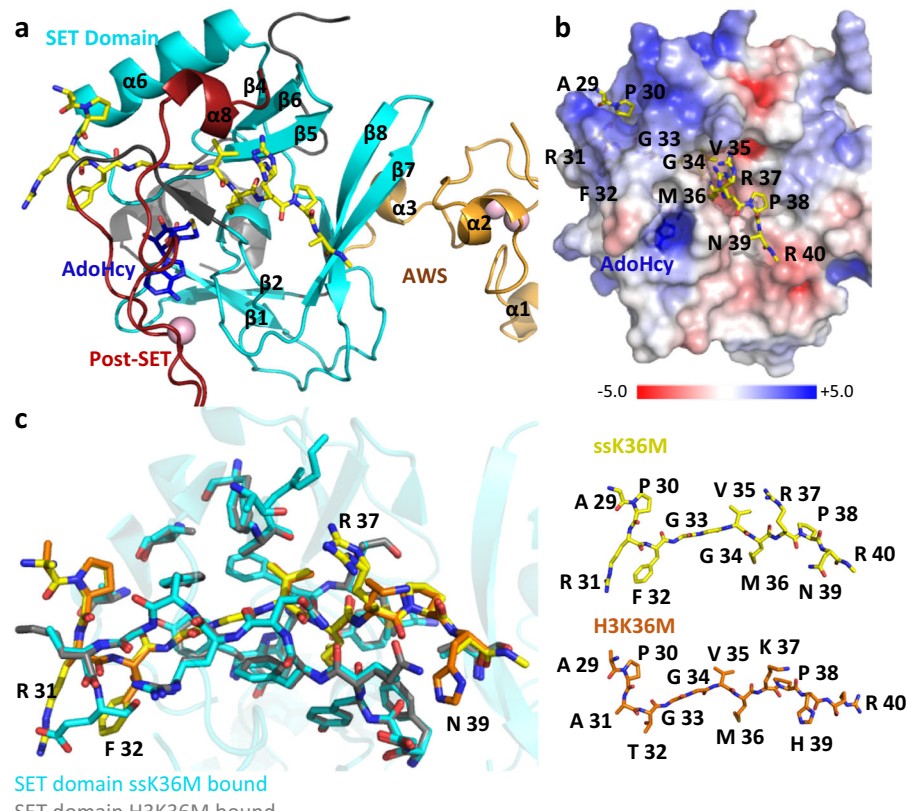

**Fig. 4 Crystal structure of the ternary complex of SETD2 bound to the ssK36M peptide and AdoHcy. a** Overall structure of SETD2 (1435–1711) bound to the ssK36M peptide and AdoHcy. SETD2 is shown as cartoon, color coded in orange for the AWS domain, cyan for the SET domain, red for the post-SET domain. The ssK36M peptide and AdoHcy are shown in yellow and blue sticks. **b** Surface representation of the protein in the same orientation as in panel A colored by electrostatic potential showing negative, positive, and neutral regions in blue, red, and white, respectively. The peptide is shown in stick. **c** Superposition of ssK36M and H3K36M bound (PDB: 5V21) SETD2 crystal structures. The protein parts are shown in cyan and gray the peptides in yellow and orange. The four amino acid substitutions of the super-substrate are labeled (R31, F32, R37, N39). The peptide conformations are shown in the right part in stick models.

substrate-binding mode were not notably different from the previously reported ternary complex with H3K36M and AdoMet (PDB 5V21)[12], as we could see from the aligned enzyme-substrate interaction surfaces of these two complexes (Fig. 4c). We use single and three letter amino acid codes for the residues of the peptide and SETD2, respectively. Residues 29-APRFGGVM-36 of ssK36M are inserted into a surface channel of the SETD2 protein and covered by the loops between α6 and β5; and post-SET and α8 (Fig. 4b) and the peptide conformation closely mimics the previously described structures of the H3K36M and H3.3 K36M peptide bound to SETD2 (PDB 5V21 and 5JJY)[11,12].

**Structural basis for the preferable ssK36 methylation**. Interaction analysis by LIGPLOT[32] based on the ssK36M-SETD2 complex structure indicated 18 hydrogen-bonds and 31 spots for hydrophobic interactions across the ssK36M-SETD2 interface, ignoring solvent atoms, but only 14 and 28 for the corresponding H3K36M structure (PDB 5V21)[12] (Supplementary Fig. 8). As expected, the majority of hydrophobic interactions occur in the SET domain channel that is occupied by the substrate residues 29 through 36. The super-substrate carries four amino acid exchanges (A31R, T32F, K37R, and H39N) with respect to the canonical H3K36 peptide. To compare the substrate interaction differences mediated by these four residues, we superimposed ssK36M and H3K36M bound (PDB 5V21) SETD2 crystal structures (Fig. 4c) by overlaying the SETD2 molecules. Although not

completely resolved by electron density, the side chain of ssK36M R31 (in its most likely conformation ttt180[33,34]) may form a salt bridge with the side chain of Glu1674 (Fig. 5a). The aromatic F32 side chain of ssK36M is inserted into a shallow pocket formed by the side chains of Glu1674 and Gln1676 making more extensive hydrophobic interactions than T32 of H3K36M (Fig. 5b). Neither the H3K36M complex structure[12] nor the present ssK36M structure clearly resolves the side chains of K37 or R37, respectively, yet in the ssK36 complex a difference density near the Ala1699 and Ala1700 carbonyl groups suggests possible hydrogen bonds with the R37 guanidinium moiety[35] (Fig. 5c). We could not infer any specific interactions involving the side chain of N39, equivalent to H3-H39, from the complex structure. In conclusion, the substituted residues in the super-substrate ssK36M interact with SETD2 mainly through the post-SET domain loop and the α6 helix of the SETD2 catalytic domain (Fig. 5d). Slotting of the F32 phenyl ring into a pocket between the Glu1674 and Gln1676 and possible interactions involving the guanidine moieties of R31 and R37 provide a plausible rationale for the enhanced methylation activity by SETD2 due to a tighter binding affinity between the super-substrate peptide and SETD2.

**H3K36 and ssK36 methylation at the protein level**. To study methylation of the super-substrate at the protein level, the ssK36 sequence, its K36M mutant (ssK36M) and the H3K36 (29–43) sequence were cloned with C-terminal GST-tag,

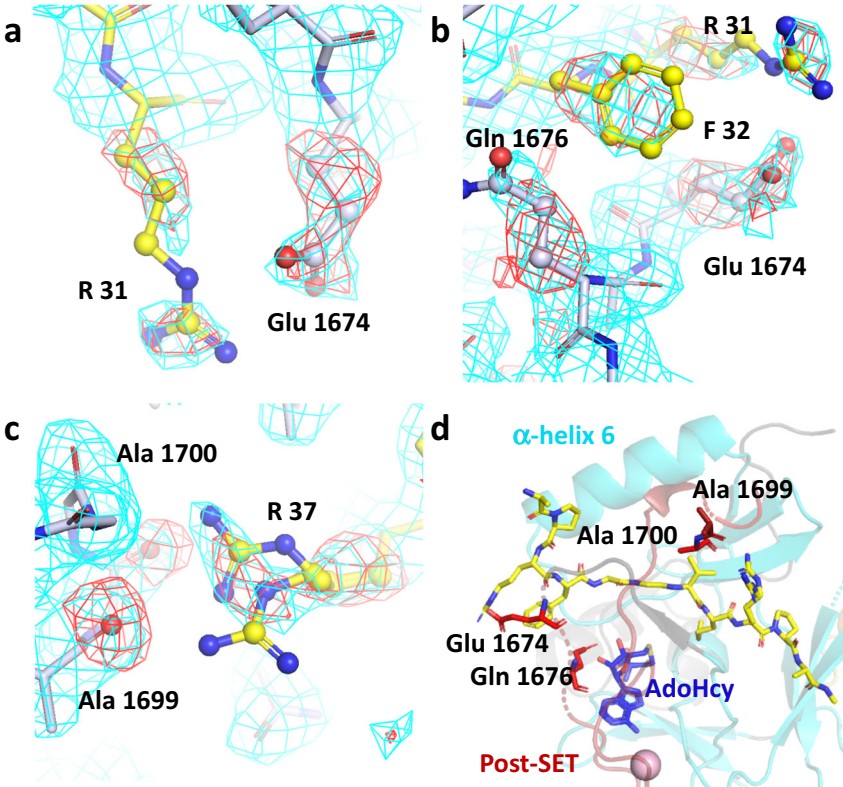

**Fig. 5 Close-up view of the interactions between the ssK36M peptide and SETD2 catalytic domain.** Details of the interaction of R31 (**a**), F32 (**b**), and R37 (**c**) with SETD2 residues. Gray and yellow sticks represent SETD2 and the ssK36M peptide. Omit mFo-DFc (red mesh, contoured with PyMOL at 3 rmsd level) and 2mFo-DFc maps (cyan, contoured at 1 rmsd level) were calculated with PHENIX. Atoms omitted for map calculation are shown as spheres. Two conformations were modeled for the side chain of ssK36M R37 to denote uncertainty in the true coordinates. **d** Overview showing that the substituted amino acids in ssK36M interact with the post-SET domain loop (red) and α6 of the SETD2 catalytic domain.

expressed and purified through glutathione sepharose beads. Afterwards, equal amounts of ssK36-GST, H3K36-GST and ssK36M-GST were used for methylation by SETD2 (Fig. 6a). Native mononucleosomes (MN) isolated from HEK293 cells were used as a positive control. We observed that the ssK36-GST protein and the MN were strongly methylated, but no signal was detected for the H3K36-GST protein. Therefore, the experiment was repeated with a longer exposure time of the film. In addition, different dilutions of ssK36-GST were loaded onto the gel to allow for a better comparison of the activities (Fig. 6b). Densitometric analysis of the band intensities revealed an about 50–65 fold higher methylation activity of SETD2 on ssK36-GST than on H3K36-GST.

For analysis of cellular methylation of the super-substrate, we first validated that an H3K36me3 antibody also detects lysine methylation in the context of the ssK36 sequence (Supplementary Fig. 9a). Next, YFP-tagged ssK36 and H3K36 (29–43) substrates were transfected into HEK293 cells together with DsRed-tagged SETD2 or empty vector expressing DsRed. The SETD2 expression (or DsRed expression in the empty vector control) was analyzed by FACS showing roughly equal expression levels (Supplementary Fig. 9b–d). The YFP-tagged ssK36 and H3K36 substrates were isolated by GFP-trap and quantified by Western Blot with anti-GFP antibody (Fig. 6c). Next, the methylation of the substrates was analyzed using the H3K36me3 antibody revealing strong methylation of the super-substrate after co-transfection with SETD2 (Fig. 6c). No methylation was observed without SETD2 co-transfection, suggesting that the endogenous SETD2 is limited. Moreover, the H3K36 sequence was not methylated even after co-transfection with SETD2, indicating that the super-substrate is

methylated much more efficiently than the H3K36 sequence in cells as well. These results were confirmed in three independent transfection series.

## Discussion

SETD2 is a PKMT initially in 2005 identified to methylate H3K36[8]. So far, lysine 36 in H3 was still the best substrate known for this enzyme at the peptide level. At the protein level, H3 methylation by SETD2 is weak, but it is strongly stimulated in the context of a nucleosome as shown here by us and previously by others[8]. Besides H3K36, two non-histone substrates of SETD2 are known, α-tubulin methylated at K40 and STAT1 methylated at K525, but methylation of these substrates was reported to be weaker than H3K36[18,19], and in our hands both were not methylated on peptide arrays. In this work, the substrate specificity profile of SETD2 was determined. Our results indicate that SETD2 is a very specific PKMT with recognition of G33 and G34, readout of hydrophobic residues at V35 and P38 and a preference for arginine at R40. The weak methylation of the α-tubulin (DGQMPSDKTIGGGDD) and STAT1 (LNMLGEKLLGPNA) peptides is in agreement with the specificity profile, because both of them contain amino acid residues that are not among the preferred ones. Based on our specificity profiles, additional non-histone peptide substrates of SETD2 were identified and 9 of them showed strong methylation at the predicted target lysine. One non-histone protein (Fibrillin 1, FBN1) was also identified as a SETD2 substrate and the target site was validated to be K666. Based on our results and literature data[18,19], FBN1 is more efficiently methylated than H3, α-tubulin and STAT1, but methylation of H3K36 in the nucleosomal context is even stronger.

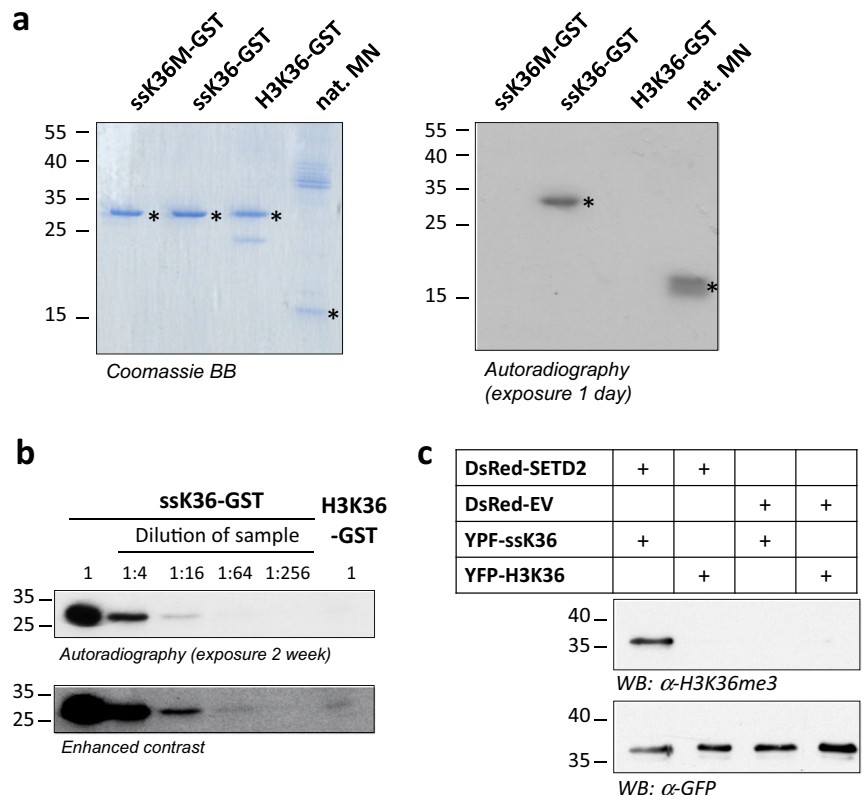

**Fig. 6 Protein methylation of the super-substrate by SETD2 in vitro and in cells. a** In vitro methylation of ssK36-GST and H3(29–43)-GST fusion proteins. The left panel shows a Coomassie BB stained SDS gel of the purified ssK36-GST and H3K36-GST fusion proteins. The ssK36 target lysine to methionine mutant (ssK36M-GST) was included as negative control and native mononucleosomes (nat. MN) as positive control. The right panel shows an autoradiography of the SETD2 methylation of the proteins. The asterisks indicate the expected sizes of the GST fusion proteins and of H3 in the mononucleosomes. **b** Comparison of the methylation of ssK36-GST and H3K36-GST. Identical amounts of both proteins were methylated by SETD2 and methylation analyzed by autoradiography. For comparison, different dilutions of the methylated ssK36-GST were loaded as indicated. **c** Cellular methylation of the super-substrate. YFP-tagged ssK36 and H3 (29–43) substrates were transfected into HEK293 cells together with DsRed-tagged SETD2 or DsRed empty vector (EV). The YFP-tagged substrates were isolated by GFP-trap and quantified by Western Blot with anti-GFP antibody. The methylation of the substrates was analyzed using an H3K36me3 antibody (see Supplementary Fig. 9 for additional controls).

Methylation of FBN1 is in agreement with the reported cellular localization of SETD2 and FBN1 (www.proteinatlas.org[36], retrieved in March 2020), because in addition to the nuclear localisation of SETD2 and extracellular localization of FBN1, cytoplasmic localization has been observed for both proteins as well.

FBN1 is a major component of the 10–12 nm diameter microfibrils of the extracellular matrix (ECM)[37]. These microfibrils are highly conserved macromolecular assemblies[38] and mutations of FBN1 cause different diseases, for example, the Marfan syndrome[39]. FBN1 contains 42 Calcium-binding EGF-like (EGFCA) domains together with EFG, EGFL and TB domains. The methylation site K666 is located between the EGFCA domain 6 (aa 613–653) and a Cys-rich TB domain (aa 670–710). K666 has already been found to be acetylated, ubiquitylated and sumoylated (Phosphosite plus[40], retrieved in May 2020) indicating that post-translational modification of this residue is possible. While further work needs to be done to confirm that FBN1 is a cellular target of SETD2 and identify the biological role of its methylation, SETD2 depletion has been associated with decreased bone matrix formation[41] and reduced interaction of endothelial cells with the surrounding cell and matrix[42]. For the other strong non-histone peptide substrates, where we could not detect protein methylation, it remains to be seen whether they are methylation substrates of SETD2. However, it is interesting to note that COMA and HMCN1 have roles in

cell-cell adhesion and extracellular matrix as well and FBN2 is a close homolog of FBN1.

Strikingly, our data revealed that the substrate specificity preferences of SETD2 differ from H3K36 at five sites; R is favored at position −5 instead of A31, F instead of T32 at position −4, R instead of K37 at position +1, N instead of H39 at position +3 and R instead of Y at +4. Based on these findings, a super-substrate peptide was designed and shown to be methylated about 290-fold better than the original H3K36 peptide. This is a striking result illustrating the power of PKMT substrate design on the basis of specificity profiles. The super-substrate fused to GST (ssK36-GST) was also methylated much more strongly than the equivalent H3K36-GST protein, confirming the distinctive methylation at the protein level. In fact, methylation of ssK36-GST was comparable to the methylation of H3 in a nucleosomal context, making ssK36-GST the best protein substrate of SETD2 that is known so far. Highly preferred methylation of the super-substrate fused to YFP by SETD2 was also confirmed in human cells.

The kinetic results with SETD2 and the super-substrate are in good agreement with the structures of SETD2 in complex with different H3K36 peptides[11,12] and the structure of SETD2 with bound super-substrate K36M inhibitor determined here. The high overall specificity of SETD2 including readout of substrate residues from P30 to R42 is in agreement with the structural data showing that the ssK36M and H3K36M peptides are deeply

engulfed by the enzyme. G33 and G34 are bound to narrow binding sites, which do not leave space for larger amino acid side chains and V35 is embedded in a hydrophobic pocket. These residues are present in the normal H3K36 sequence and in the super-substrate sequence. The 3.5-fold improved inhibition constant of ssK36M when compared with the H3K36M inhibitor observed in the steady-state inhibition kinetics can be explained by possible additional interactions observed in the SETD2-ssK36M complex structure; ssK36M R31 could interact with Glu1674, F32 is sandwiched into a pocket generated by Glu1674 and Gln1676, and R37 could interact with the peptide carbonyl groups of Ala 1699 and Ala1700.

As shown in the kinetic analysis, the increased methylation of the super-substrate is mainly due to an increase in the rate of catalysis and only to a smaller degree to improved binding. This observation suggests that conformational changes occur after substrate binding which are leading to the increased catalytic rate of SETD2. This conclusion is also in agreement with the finding that methylation of H3K36 in a nucleosomal context is much faster than methylation of the H3 protein. To identify parts of the structure that might undergo conformational changes, we superimposed the SETD2-ssK36M complex with the H3K9 complex of the highly active DIM5 H3K9 PKMT (pdb 1PEG)[43]. While the conserved SET domains of both proteins superimposed well, the C-terminal part of α-helix 6 of SETD2 next to R31 and F32 is shifted leading to a more open peptide binding cleft (Fig. 7a). Interestingly, movement of the same helix has been linked to regulation of the activity of the MLL1 (KMT2A) PKMT previously[44]. In addition, the C-terminal part of the bound ssK36M peptide including R37 and N39 is shifted when compared with the conformation observed in the DIM5-H3K9 complex (Fig. 7b). Hence, our data suggest that the amino acids exchanged in the super-substrate either allow for the formation of additional contacts to the enzyme or they influence the conformational dynamics and by this they lead to the massive rate enhancement of methylation of the super-substrate.

In conclusion, our data show the importance of mapping the substrate specificity of PKMTs, because based on specificity profiles, new substrates can be identified and mechanistic insights can be gained as demonstrated here for SETD2. The discovery of novel non-histone substrates of PKMTs is particularly important to gain more insights into their biological roles. Lysine methylation can cause downstream effects, like changing the protein stability or modulation of protein/protein interaction, which then

lead to biological outcomes. The most surprising result of our work is the design of an artificial SETD2 substrate that is methylated about 290-fold faster than H3K36, the best substrate of this enzyme known so far. These data illustrate the great potential in substrate sequence design to increase the specific activity of PKMTs. Strikingly, the SETD2 super-substrate sequence identified here does not exist in the human proteome. It will be interesting to find out the reason for this unexpected specificity of this important PKMT. One evolutionary scenario to explain this surprising finding could be that SETD2 has been simultaneously optimized in evolution for the methylation of more than one protein substrate leading to a "mixed" adaptation. Then, the amino acid sequence that interacts best with the binding site may differ from all individual substrates and it may even not correspond to an existing protein sequence as observed here for SETD2.

## Methods

**Cloning, expression, and purification**. The His$_6$-tagged expression construct of SETD2 catalytic SET domain (amino acids 1347–1711, UniProt No: Q9BYW2) was kindly provided by Dr. Masoud Vedadi[9]. The expression and the purification of the His$_6$-tagged SETD2 protein was conducted as described previously[30]. The domain constructs of the non-histone substrates were cloned by PCR using cDNA isolated from HEK293 cells as template. The domain boundaries of the non-histone substrates were designed with the Scooby domain prediction tool (http://www.ibi.vu.nl/programs/scoobywww/)[45]. For the introduction of the target lysine mutation to arginine, site-directed mutagenesis was used[46]. H3K36 (29–43)-GST was cloned by Gibson-Assembly from an H3 (1–60) plasmid constructed by Dr. Rebekka Mauser. The ssK36-GST (29–43) and the ssK36M-GST (29–43) constructs were cloned by site-directed mutagenesis method using the H3K36 (29–43)-GST plasmid as template. As protein expression vector for all constructs, pGEX-6p2 was used. All cloning steps were confirmed by sequencing. The protein expression and purification of the GST-tagged non-histone substrate proteins and the H3K36 (29–43)-GST variant proteins were performed as described previously[47]. For mammalian expression, the coding sequence of SETD2 (amino acids 1347–1711, UniProt No: Q9BYW2) was cloned into the pMulti-sgRNA-LacZ-DsRed (Addgene) by Gibson-Assembly. The H3K36 (29–43) and ssK36 (29–43) were cloned into the pEYFP-C1 vector (Clontech, Palo Alto, CA, USA) using synthetic DNA sequences as inserts.

**Cell culture, transfection, and immunoprecipitation**. HEK293 cells were grown in Dulbecco's Modified Eagle's Medium (Sigma) supplemented with 5% fetal bovine serum, penicillin/streptomycin, and L-glutamine (Sigma). The DsRed fused SETD2 was co-transfected with YFP-fused H3K36 (29–43) or ssK36 (29–43) into cells using polyethylenamine (Promega, according to the manufacturer's instructions). Transfection efficiency and expression of the transgenes of HEK293 cells was evaluated by flow cytometry (MACSQuant VYB, Miltenyi Biotec) using the DsRed reporter. Data analysis was performed using the FlowJo software (Treestar). For methylation analysis, the cells were washed with PBS buffer (Sigma) 72 h after transfection and harvested by centrifuging at 525 × g for 5 min. The YFP-fused

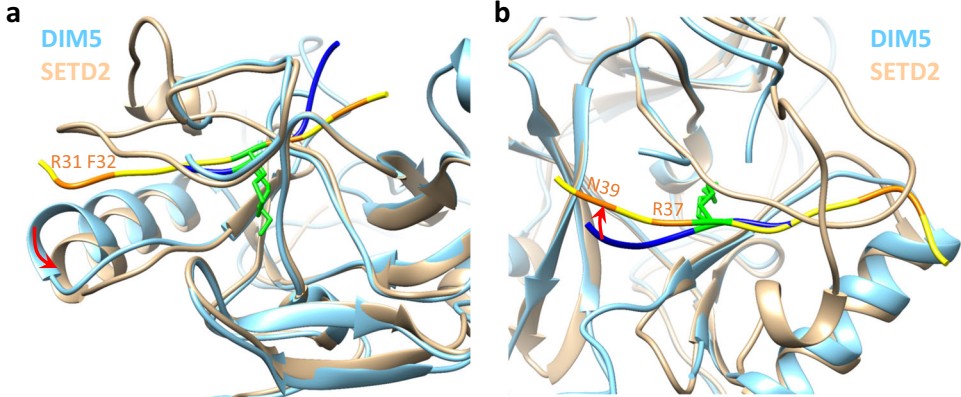

**Fig. 7 Superposition of the SETD2-ssK36M with the structure of DIM5 (pdb 1PEG)[43].** DIM5 is shown in light blue, SETD2 in tan. The H3K9 peptide bound to DIM5 is shown in dark blue with K36 in green. The ssK36M bound to SETD2 is shown in yellow with the substituted positions in orange and M36 in green. **a** While the conserved SET domains of SETD2 and DIM5 superimpose well, the C-terminal part of α-helix 6 next to R31 and F32 is shifted in SETD2 leading to a more open peptide binding cleft. **b** The C-terminal part of the bound ssK36M peptide including R37 and N39 is shifted when compared with the conformation observed in the DIM5-H3K9 complex.

H3K36 (29–43) or ssK36 (29–43) protein was purified from mammalian cell extracts using GFP-Trap A beads (Chromotek) following the manufacturer's instructions. The samples were heated to 95 °C for 5 min in SDS-gel loading buffer and resolved by 16% SDS-PAGE. Analysis was performed by Western Blot using as primary antibody H3K36me3 (ab9050, lot. GR3257952-1) or GFP antibody (Clontech, lot. 1404005) and as secondary antibody anti-rabbit HRP (GE Healthcare, lot. 9739638).

**Peptide array methylation**. Peptide arrays containing fifteen amino acid long peptides were synthesized using the SPOT synthesis method[48] with an Autospot peptide array synthesizer (Intavis AG, Köln, Germany). After synthesis, the obtained membranes were pre-incubated in methylation buffer (20 mM Tris/HCl pH 9, 0.01% Triton X-100, 10 mM DTT, 1.5 mM MgCl₂) for 5 min on a shaker. Thereafter, the membranes were incubated in methylation buffer supplemented with 0.76 µM radioactively labeled AdoMet (PerkinElmer) and 3–6 µM SETD2 enzyme for 60 min on a shaker. After methylation, the membranes were washed 5 times for 5 min in wash buffer (100 mM NH₄HCO₃, 1% SDS) and incubated in amplify NAMP100V (GE Healthcare) for 5 min. This was followed by the exposure of the membranes to a hyperfilm™ high performance autoradiography (GE Healthcare) film at −80 °C in the dark. The films were developed with an Optimax Typ TR machine after different exposue times. For quantification the signal intensities were measured with the Phoretix™ software and analyzed with Microsoft Excel.

**Peptide methylation assay**. The H3K36 (27–43) and the H3K36M (27–43) peptides were purchased from Intavis AG (Köln, Germany) and the ssK36 peptide (29–43) and the ssK36M peptide (29–43) were from Innovative Peptide Solutions (Berlin, Germany). If not indicated otherwise, peptide methylation reactions were performed using 20 µM peptide and 6 µM SETD2 in 20 µl methylation buffer supplemented with 0.76 µM radioactive labeled AdoMet (PerkinElmer) for 3 h at 25 °C. The reactions were halted by the addition of Tricine-SDS-PAGE loading buffer and incubation for 5 min at 95 °C. Afterward the samples were separated by Tricine-SDS-PAGE, which was followed by the incubation of the gel in amplify NAMP100V (GE Healthcare) for 45 min on a shaker and drying of the gel for 2 h at 55 °C in vacuum. The signals of the transferred radioactive labeled methyl groups were detected by autoradiography using a hyperfilm™ high performance autoradiography film (GE Healthcare) at −80 °C in the dark. The film was developed with an Optimax Typ TR machine after different exposure times.

Time courses of ssK36 methylation were analyzed by fitting to an exponential reaction progress curve (Eq. 1). Initial rates of single turnover reactions were determined using the first derivative of eq. 1 at $t = 0$.

$$\text{Signal} = \text{BL} + \text{F} \times \left(1 - \exp^{-k_{st} \times t}\right)$$

Equation 1: Exponential reaction progress curve used to fit single turnover methylation kinetics (BL, baseline; F, Signal factor; $t$, time; $k_{st}$, single turnover rate constant).

Multiple turnover kinetics were determined using variable peptide concentrations (20–320 µM). Methylation levels were fitted to the Michaelis Menten model (equation 2).

$$v = \frac{v_{max}}{c_s + K_M}$$

Equation 2: Michaelis–Menten model used to fit multiple turnover kinetics ($v$, reaction rate; $c_S$, concentration of the substrate peptide; $K_M$, Michaelis-Menten constant; $v_{max}$, turnover number).

**Methylation assays with K36M peptide inhibitors**. The ssK36 (29–43) peptide was used in a concentration of 20 µM. The H3K36M (27–43) and the ssK36M (29–43) peptide inhibitors were used in a concentration ranging from 0 to 320 µM. The peptide methylation experiments were performed as described above. The results were globally fitted by least squares fit with Microsoft Excel Solver module under standard settings (GRL non-linear) to equation 3, which describes a competitive inhibition steady-state model.

$$v = \frac{v_{max}}{c_s + K_M\left(1 + \frac{c_I}{K_I}\right)}$$

Equation 3: Competitive inhibition model used to fit multiple turnover inhibition kinetics ($v$, reaction rate; $c_S$, concentration of the substrate peptide; $c_I$, concentration of the inhibitor peptide; $K_M$, Michaelis–Menten constant; $K_I$, inhibition constant).

**Protein methylation assay**. Protein methylation reactions were performed in 40 µl methylation buffer containing 0.76 µM radioactive labeled AdoMet (PerkinElmer), approximately 20 µg of substrate protein and 6 µM SETD2 enzyme (if not otherwise indicated) for 3 h at 25 °C. Thereafter, the reaction was halted by the addition of SDS-PAGE loading buffer and incubation of the samples for 5 min at 95 °C. Afterward, the samples were separated by a 16% SDS-PAGE. The SDS-gel was then incubated in amplify NAMP100V (GE Healthcare) for 45 min on a shaker and dried for 90 min at 65 °C in vacuum. The radioactive methylation signal was

detected by a hyperfilm™ high performance autoradiography film (GE Healthcare) at −80 °C in the dark. The films were developed with an Optimax Typ TR machine after different periods of time. The recombinant H3.1 protein (M2503S) was purchased from NEB.

**Native mononucleosome isolation**. HEK293 cells were grown in Dulbecco's Modified Eagle's Medium (Sigma) supplemented with 5% fetal bovine serum, penicillin/streptomycin, and L-glutamine (Sigma). HEK293 cell pellets was thawed on ice and afterward resuspended in ice cold resuspension buffer (RB) (15 mM Tris/HCl pH 8, 15 mM NaCl, 60 mM KCl, 250 mM sucrose, 5 mM MgCl₂, 1 mM CaCl₂, 1 mM DTT, 200 µM PMSF). Thereafter, the cells were centrifuged at 530 g for 5 min at 4 °C and the supernatant was discarded. These two steps were repeated. Afterward, the cells were resuspended in a 1:1 mixture of ice cold RB and lysis buffer (15 mM Tris/HCl pH 8, 15 mM NaCl, 60 mM KCl, 250 mM sucrose, 5 mM MgCl₂, 1 mM CaCl₂, 1 mM DTT, 200 µM PMSF, 0.6% NP-040) and lysed for 10 min on ice. Then, the cells were centrifuged at 530 × g, for 5 min at 4 °C and the supernatant was discarded. The nuclei were resuspended in RB and transferred into a new chilled eppendorf tube. The nuclei were centrifuged at 600 × g, for 5 min at 4 °C and the supernatant was discarded. This step was followed by a washing step of the nuclei with RB and another centrifugation of the nuclei at 600 × g, for 5 min at 4 °C. Thereafter, the nuclei were resuspended in RB and the DNA concentration was measured with the NanoDrop spectrometer. After this step, the nuclei were digested with micrococcal nuclease (1 µl per 300–350 ng/µl) (NEB, M0247S) at 37 °C for 10 min. The reaction was stopped by the addition of EGTA-EDTA buffer (1/10 of the volume) and this was followed by centrifugation at maximum speed, for 10 min at 4 °C. Thereafter, the supernatant was transferred into a new chilled eppendorf tube and the DNA concentration was measured with the NanoDrop spectrometer.

**Histone purification**. Individual histone proteins were overexpressed in BL21 DE3 codon plus cells. The cells were grown to an OD₆₀₀ of 0.6, induced for 4 h with 1 mM isopropyl-β. -D-thiogalactopyranoside (IPTG), and harvested by centrifugation (5000 × g, 20 min). The cell pellet was resuspended in lysis buffer containing 40 mM sodium acetate pH 7.5, 1 mM EDTA, 10 mM lysine, 5 mM β-mercaptoethanol, 6 M urea and 200 mM NaCl, and disrupted by sonication (5 min of sonication, with 40% power). The lysate was cleared by centrifugation (20,000 × g, 1 h) and passed through a 0.45 µm syringe filter to remove particulate matter. The solution was then passed over a 5 ml SP HP cation exchange column (GE Healthcare) which was equilibrated with lysis buffer. After washing with additional 5 column volumes (CV) of lysis buffer, the histone proteins were eluted using a salt gradient from 200 to 800 mM NaCl. Fractions containing pure histone proteins were dialyzed against water and subsequently lyophilized and stored at 4 °C.

**Octamer reconstitution**. Lyophilized aliquots of histone proteins were dissolved in unfolding buffer containing 7 M guanidiniumchloride, 20 mM Tris/HCl pH 7.5, and 5 mM DTT for 30 min. Concentration was determined by spectrophotometry and the proteins were pooled according to a molar ratio of 1 (H3, H4) to 1.2 (H2A, H2B) to prevent the formation of free (H3-H4)₂ tetramers. The mixture was dialyzed against refolding buffer containing 2 M NaCl, 10 mM Tris/HCl pH 7.5, 1 mM EDTA, and 5 mM β-mercaptoethanol. Fully reconstituted histone octamers were separated from (H2A-H2B) dimers by size exclusion chromatography using a Superdex 200PG column (GE Healthcare) equilibrated with refolding buffer. Fractions containing pure histone octamers were concentrated tenfold using Amicon spin filters (Merck), flash frozen in liquid nitrogen and stored at −80 °C.

**Nucleosome reconstitution**. A 240 bp sequence containing the Widom 601 sequence was amplified by PCR and purified using the Nucleospin PCR cleanup kit (Macherey-Nagel). Purified DNA and histone octamers were mixed in equimolar ratio together with 5 M NaCl to a final DNA concentration of 0.5 mg/ml and 2 M NaCl. The sample was dialyzed against high salt buffer (2 M NaCl, 10 mM Tris/HCl pH 7.5, 1 mM EDTA, 1 mM DTT) which was continuously replaced by low salt buffer (same with 250 mM NaCl) over a time of 24 h. Successful nucleosome assembly was confirmed by electrophoretic mobility shift assay (EMSA), aliquots were flash frozen in liquid nitrogen and stored at −80 °C.

**Protein purification and crystallization**. Wild-type human SETD2 catalytic domain (residues 1435–1711) was expressed in *Escherichia coli* (*E. coli*) cells and purified in the form of His-tagged proteins as previously described[31]. Briefly, N-terminal His-tagged plasmids were transformed into *E. coli* (DE3) RIL cells and, protein expression was induced by 0.5 mM IPTG at 14 °C for 16 h. Proteins were purified using Ni-NTA affinity agarose resin (QIAGEN), followed by running through cation-exchange (HiTrap SP HP 5 mL, GE Healthcare) and size-exclusion (Superdex 200, GE Healthcare) columns in a final buffer of 25 mM Tris/HCl (pH 8.0), 200 mM NaCl. For ternary complex crystal growth, the SETD2 protein (10 mg/ml) was incubated with AdoMet and the ssK36M peptide at a molar ratio of 1:10:5. The crystallization condition was optimized based on a previously reported condition for the crystal growth of a SETD2 ternary complex[11]. Crystals were obtained via the hanging drop method by mixing 1 µL of the protein complex and 1 µL of reservoir solution (0.2 M KSCN, 0.1 M Tris/HCl pH 8.5, and 26% PEG

3350) at 18 °C. Next, the crystals were mounted and soaked in a cryo-protectant composed of reservoir solution supplemented with 10–20% glycerol and were flash-frozen in liquid nitrogen for data collection.

**Data collection and structure refinement**. Diffraction experiments were conducted initially on a rotating copper anode and, later, at Advanced Photon Source beam line 24-ID-E. Diffraction images were processed with XDS[49] and AIMLESS[50]. Coordinates from isomorphous PDB entry 5JLB chain A were refined against these diffraction data using DIMPLE/REFMAC[51]. The model was interactively rebuilt in COOT[34] and further refined in REFMAC, AUTOBUSTER (BUSTER. Cambridge, United Kingdom: Global Phasing Ltd.) and PHENIX[52]. The latest round of refinement was performed with BUSTER against the weaker, but earlier collected, copper anode data, which we preferred due to its lower radiation load and concerns about radiation-induced decarboxylation of glutamate residues[53]. Two conformations were modeled for the side chain of ssK36M R37 to denote uncertainty in the true coordinates, not to denote the observation of any specific conformation. We failed to fit a favored arginine rotamer to the R37 difference electron density. Diffraction data were deposited at the Integrated Resource for Reproducibility in Macromolecular Crystallography (proteindiffraction.org) as entry SETD2_HR606-6vdb (https://proteindiffraction.org/project/SETD2_HR606_6vdb/). The current atomic model has been deposited in the Protein Data Bank (https://www.rcsb.org/) as entry 6VDB (https://www.rcsb.org/structure/6VDB). The PDB2PQR webserver has been used to prepare some of the illustrations[54].

**Statistics and reproducibility**. All experiments were conducted in replicates as indicated.

**Reporting summary**. Further information on research design is available in the Nature Research Reporting Summary linked to this article.

## Data availability

The datasets generated during the study are available from the corresponding authors upon request. Diffraction data were deposited at the Integrated Resource for Reproducibility in Macromolecular Crystallography (proteindiffraction.org) as entry SETD2_HR606-6vdb. The current atomic model has been deposited in the Protein Data Bank (https://www.rcsb.org/) as entry 6VDB. All biochemical data generated or analyzed during this study are included in the published article and its supplementary files. Source data underlying plots shown in figures and full blots are provided in Supplementary Data 1.

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

## Acknowledgements

We thank Dr. Masoud Vedadi for the gift of the His$_6$-tagged SETD2 bacterial expression construct and Dr. Rebekka Mauser for the H3 (1-60) plasmid construct. This work has been supported by the DFG grant JE 252/7–4 (AJ), and the GERLS scholarship program (MSK) funding program number 57311832 by the German Academic Exchange Service (DAAD) and the Egyptian Ministry of Higher Education. The SGC is a registered charity (number 1097737) that receives funds from AbbVie, Bayer Pharma AG, Boehringer Ingelheim, Canada Foundation for Innovation, Eshelman Institute for Innovation, Genome Canada through Ontario Genomics Institute [OGI-055], Innovative Medicines Initiative (EU/EFPIA) [ULTRA-DD grant no. 115766], Janssen, Merck KGaA, Darmstadt, Germany, MSD, Novartis Pharma AG, Pfizer, São Paulo Research Foundation-FAPESP, Takeda, and Wellcome. This work is based upon research conducted at the Northeastern Collaborative Access Team beamlines, which are funded by the National Institute of General Medical Sciences from the National Institutes of Health (P30 GM124165). The Eiger 16 M detector on 24-ID-E beam line is funded by a NIH-ORIP HEI grant (S10OD021527). This research used resources of the Advanced Photon Source, a U.S. Department of Energy (DOE) Office of Science User Facility operated for the DOE Office of Science by Argonne National Laboratory under Contract No. DE-AC02-06CH11357. Open access funding provided by Projekt DEAL.

## Author contributions

M.K.S., S.W., J.M., and A.J. designed the experiments. M.K.S and M.S.K. conducted the biochemical experiments with the help of J.L. and S.W. S.B. conducted the structural studies with help of W.T. A.B. prepared the reconstituted mononucleosomes. M.K.S., S.W., M.S.K., J.M., and A.J. did the data analysis and interpretation. M.K.S., S.W., J.M., and A.J. drafted the manuscript. All authors contributed to the editing of the manuscript and approved its final version.

## Competing interests

The authors declare no competing interests.
