## [Peer Review File · Communications Biology]

Reviewers' comments:

Reviewer #1 (Remarks to the Author):

The manuscript by Schuhmacher et al reported a comprehensive study of the sequence specificity of a lysine methyltransferase SETD2, based on the reported substrate H3K36. Such analysis allowed the authors to identify a novel protein substrate FBN1 as well as an optimal super substrate ssK36. The authors showed evidence that the ssK36 can be methylated much more efficiently than wild-type H3K36 both in vitro and in cells. Furthermore, structural analysis of ssK36M in complex with SETD2 provided possible explanation of the superior catalytic efficiency. Overall, the manuscript is well organized and clearly described. I only have a few minor comments.

1. For the structural description, the authors should provide an overall omit map of the peptide (a stereo view is better) to show the quality of the model building.
2. The authors claimed that the 1000-fold increase in methylation rate comes from an increase in kcat. It would be better if the authors can design experiments to determine the quantitative number of kcat for ssK36 and H3K36.
3. A follow-up suggestion: the authors may consider moving the structural analysis of the conformational changes from the discussion to the results part.
4. In the discussion part, the authors discuss about the cellular localization of SETD2 and FBN1. But how can a nuclear or cytoplasmic enzyme SETD2 catalyze an extracellular ECM protein FBN1?
5. It is interesting to see that ssK36M can serve as an inhibitor in vitro. Is the ssK36M peptide specifically inhibit SETD2 but not other methyltransferases? Can over-expression of ssK36M peptide in cells reduce the methylation level of endogenous H3K36 in nucleosomes?

Reviewer #2 (Remarks to the Author):

In this manuscript, Schuhmacher et al. present the first substrate specificity analysis of the lysine methyltransferase SETD2. The authors successfully identify a novel "super" substrate that is methylated more robustly at the peptide level than previously reported non-histone substrates and histone substrates. Overall this is a solid body of work that warrants publishing; however, enthusiasm for this manuscript is slightly diminished by the following concerns that should be addressed prior to publication:

1. The authors report error analysis of their peptide array results (Supplemental Figure 2) and conclude that there is a "high reproducibility of the results." However, while the authors correctly state that "most spots had an SD smaller than +/- 10%," rough estimation from the error analysis shows that ~110/280 and ~80/280 spots had greater than 10% error. The authors should quantify from the replicates and report a bar graph with SD in the supplement. This will allow readers to understand which spots are more prone to error. As the data is currently reported, it is impossible to determine if the spots that can sometimes have a deviation of 30-40% between only two technical repeats are essential for determining the novel substrates.
2. The authors state on page 6, line 187, " a second substrate specificity analysis was performed (Figure 1 D-F and Suppl. Figure 2B). In this experiment, only one amino acid change led to higher

methylation efficiency than the template substrate (I instead of V at position -1).” The authors should clarify this statement, as it appears that A29 has several substitutions that improve methylation, R37 has many substitutions that improve methylation, and P38 has positions that are better or the same.

3. On page ten, lines 362, the authors state “Based on our data, FBN1 is the most efficient natural protein substrate of SETD2 that can be more efficiently methylated than H3K36, alpha-tubulin and STAT1.” This is an overstatement for several reasons. First, the only data showing methylation of FBN1 does not directly compare to histone H3. Secondly, the authors never compare to STAT1 and alpha-tubulin at the protein level. Finally, there is data that directly contradicts the notion that FBN1 is a more efficient substrate than histone H3. In Supplemental Figure 5, the ssK36-GST is methylated more robustly than FBN1, but in Figure 6, this same ssK36-GST is methylated at a level similar to the native H3 in the nucleosome context (lanes 2 vs 4). The authors need to run additional experiments directly comparing methylation of FBN1, H3, STAT1, and alpha-tubulin in order to make these claims.

Sequence specificity analysis of the SETD2 protein lysine methyltransferase leads to the discovery of a SETD2 super-substrate COMMSBIO-19-1945-T

Reply to the reviewers' comments:

General comment: We like to thank both reviewers for critical reading of our manuscript and their overall positive assessment. We very much appreciate the insightful technical comments of the reviewers, which helped us to improve our manuscript further. Based on them, the following additional experiments were conducted:

- Repetitions of the peptide and protein methylation experiments with the ssK36 and H3K36 substrates and improved quantitative analysis
- Measurement of time courses of peptide methylation with the ssK36 and H3 peptides to determine exact methylation rate constants
- Methylation experiments of ssK36 using NSD1, NSD2 and SUV39H2
- Inhibition experiments with ssK36M and SUV39H2
- Expression of YFP tagged ssK36M in cells and investigation of the potential inhibition of SETD2 mediated histone methylation in cells
- Peptide array methylation of ssK36 variant peptides with NSD1 and NSD2

Finally, these experiments provided interesting new data, which were either included in the manuscript or they are described below. During the revision, we decided to tone down the evolutionary model to explain that proteins containing the optimized target sequences of SETD2 do not exist in an evolutionary perspective, because we realized that it is rather speculative. Moreover, the writing was improved at several places. We hope that the reviewers can now support the publication of our revised manuscript.

“Reviewer #1 (Remarks to the Author):

The manuscript by Schuhmacher et al reported a comprehensive study of the sequence specificity of a lysine methyltransferase SETD2, based on the reported substrate H3K36. Such analysis allowed the authors to identify a novel protein substrate FBN1 as well as an optimal super-substrate ssK36. The authors showed evidence that the ssK36 can be methylated much more efficiently than wild-type H3K36 both in vitro and in cells. Furthermore, structural analysis of ssK36M in complex with SETD2 provided possible explanation of the superior catalytic efficiency. Overall, the manuscript is well organized and clearly described. I only have a few minor comments.

1. For the structural description, the authors should provide an overall omit map of the peptide (a stereo view is better) to show the quality of the model building.

Reply: In the revision, we provided a simulated annealing omit map with the whole peptide omitted in Suppl. Fig. 7, which has been calculated with the Phenix software.

2. The authors claimed that the 1000-fold increase in methylation rate comes from an increase in k_{cat} . It would be better if the authors can design experiments to determine the quantitative number of k_{cat} for ssK36 and H3K36.”

Reply: The reviewer asks for more quantitative data and this point is well taken. First, we have reanalyzed the peptide spot methylation experiments more quantitatively by considering

the spot intensities and sizes. This analysis revealed that the ssK36 substrate is methylated 70 ± 10 (SD) fold times faster on peptide SPOT arrays. Next, we have repeated the methylation experiments with the ssK36 and H3K36 peptide substrates in solution and improved the quantitative readout by using appropriate dilution series. This allowed us to compare bands with similar signal intensity in the x-ray films. Based on this we now indicate that the ssK36 peptide methylation is 50 ± 4 (SD) times faster. Using the same approach, the methylation of the GST-tagged ssK36 and H3K36 protein substrates was quantified as well, leading to very similar estimates (50-65 fold increased methylation of ssK36).

To determine the k_{cat} value as requested, we attempted to carry out kinetics at higher peptide concentrations, but observed substrate inhibition and therefore were unable to determine the k_{cat} directly. As a consequence of this we have toned down the argumentation with k_{cat} . In order to obtain more quantitative data about rate constants of methylation, we conducted several additional time course experiments with the ssK36 and H3K36 peptides. First, we measured the time course of ssK36 methylation under single turnover conditions allowing us to determine the single turnover rate constant of ssK36 methylation to be 4.1 h^{-1} under these conditions, which is a value in the expected range for enzymes of this type. However, H3K36 methylation was too weak to be quantified reliably under these conditions. We therefore moved to multiple turnover time course experiments, which are allowing to load more peptide on the gel. The initial slope of these reactions with ssK36 and H3K36 could be directly compared on the same gel allowing us to conclude that the ssK36 peptide substrate is methylated 290-fold faster than the H3K36 substrate. The text was adjusted to describe these findings and the data are shown in Fig. 2C, 2D, and Suppl. Fig. 3D of the revised manuscript.

“3. A follow-up suggestion: the authors may consider moving the structural analysis of the conformational changes from the discussion to the results part.”

Reply: As this part of the manuscript mainly deals with the interpretation of the structural findings in the context of the original H3 and super-substrate sequence, we respectfully request to leave it in the discussion section.

“4. In the discussion part, the authors discuss about the cellular localization of SETD2 and FBN1. But how can a nuclear or cytoplasmic enzyme SETD2 catalyze an extracellular ECM protein FBN1?”

Reply: This is an interesting question. Extracellular proteins are synthesized inside the cell and they can be modified co-translationally or before the transfer into the ER. In case of FBN1 in addition to the secreted protein, a prominent cytoplasmic localization has been reported (www.proteinatlas.org), suggesting that the cytoplasmic residence time is rather long. Moreover, K666 of FBN1 (the identified target residue of SETD2) has been shown to carry other PTMs (acetylation, ubiquitylation and sumoylation), which also documents accessibility to cytoplasmic modification enzymes. These arguments have been added to the discussion part of our manuscript.

“5. It is interesting to see that ssK36M can serve as an inhibitor in vitro. Is the ssK36M peptide specifically inhibit SETD2 but not other methyltransferases?”

Reply: Thank you for this interesting question. We first have confirmed that the super-substrate peptide is not methylated by SUV39H2 and K36M does not inhibit SUV39H2 (Figure R1).

Figure R1: Lack of methylation of ssK36 by SUV39H2 and lack of inhibition of SUV39H2 by ssK36M.

Unexpectedly, similar studies with NSD1 and NSD2 indicated that the super-substrate indeed is a better substrate than H3K36 for NSD1 and NSD2 as well. We, therefore, repeated the peptide array used for the development of the SETD2 super-substrate, which is shown in Figure 2A of our manuscript for NSD1 and NSD2. As shown in Figure R2, after methylation with NSD1 and NSD2, all spots containing H39N showed an increased methylation indicating that this mutation specifically increases the activity of these two enzymes. In contrast, with SETD2 we observed a stepwise improvement of activity with increasing number of mutations starting with T32F, over A31R/T32F, A31R/T32F/K37R finally to A31R/T32F/K37R combined with H39N/K. Importantly, in case of SETD2 H39N only has an effect in the context of the other mutations in ssK36. We conclude that the ssK36 substrate is specifically optimized for SETD2 as expected given the workflow of our experiments. The finding that H39N increases the activity of NSD1 and NSD2 is interesting and deserves additional work. It is in line with our previous finding where we showed that H3K36 is not the ideal substrate of NSD1 and identified that H1.5 K168 is a better NSD1 substrate than H3K36 (Kudithipudi et al., 2014, Chem Biol. 21:226-37).

Spot No.	Mutation(s)	Sequence	Methylation by SETD2	Methylation by NSD1	Methylation by NSD2
A 1	-	APATGGVKKPHRYRP			
A 2	K36A	APATGGVAKPHRYRP			
A 3	A31R	APRTGGVKKPHRYRP			
A 4	A31G	APGTGGVKKPHRYRP			
A 5	T32R	APARGGGVKKPHRYRP			
A 6	T32F	APAFGGVKKPHRYRP	+		
A 7	T32Y	APAYGGVKKPHRYRP	+		
A 8	K37R	APATGGVKRPHRYRP		+	
A 9	K37H	APATGGVKHPHRYRP			
A 10	K37I	APATGGVKIPHRYRP			
A 11	K37L	APATGGVKLPHRYRP			
A 12	K37F	APATGGVKFPHRYRP			
A 13	K37Y	APATGGVKYPHYRP			
A 14	K37V	APATGGVKVPHRYRP		+	
A 15	H39R	APATGGVKKPHRYRP			
A 16	H39N	APATGGVKKPNRYRP		++	++
A 17	H39G	APATGGVKKPGRYRP		+	
A 18	H39K	APATGGVKKPKRYRP			
A 19	H39Y	APATGGVKKPYRYRP			
A 20	A31R, K37R	APRTGGVKRPHRYRP		+	
B 1	A31G, K37R	APGTGGVKRPHRYRP	+		
B 2	A31R, T32R	APRRTGGVKKPHRYRP			
B 3	A31R, T32F	APRFGGVKKPHRYRP	++		
B 4	A31R, T32Y	APRYGGVKKPHRYRP	++		
B 5	T32R	APARGGGVKKPHRYRP			
B 6	A31R, T32F, K37R	APRFGGVKRPHRYRP	++		
B 7	A31R, T32Y, K37R	APRYGGVKRPHRYRP	++		
B 8	T32R	APARGGGVKKPHRYRP			
B 9	A31R, T32F, K37Y, H39K	APRFGGVKYPKRYRP	+++		
B 10	A31R, T32F, K37R, H39N	APRFGGVKRPNRYRP	+++	++	++
B 11	A31R, T32F, K37F	APRFGGVKFPHRYRP	++		
B 12	A31R, T32F, K37Y, H39K, Y41R	APRFGGVKYPKRRRP	+++		
B 13	A31R, T32F, K37R, H39N, Y41R	APRFGGVKRPNRRRP	+++	++	++
B 14	A31R, T32F, K37F, Y41R	APRFGGVKFPHRRRP	+++		
B 15	-	APATGGVKKPHRYRP		+	
B 16	K36A	APATGGVAKPHRYRP			

Figure R2: Peptide array methylation experiments as shown in Fig. 2A of the manuscript conducted with NSD1 and NSD2. The SETD2 panel is taken from the manuscript for comparison. The table indicates the peptide compositions. Note, the gradual increase of methylation with increasing number of mutations in the case of SETD2. In case of NSD1 and NSD2, increased methylation is solely due to the H39N mutation, which has a weaker influence in SETD2.

“Can over-expression of ssK36M peptide in cells reduce the methylation level of endogenous H3K36 in nucleosomes?”

Reply: This is a very interesting question as well. We have expressed the super-substrate fused to YFP in HEK293 cells after transient transfection (see Figure R3), extracted histones after two days and detected the global H3K36me3 level by Western blot. Unfortunately, no reduction in H3K36me3 levels was detectable (see Figure R4). However, we would not like to include these negative data into the manuscript, because we do not know why the experiment failed (insufficient time of expression of the inhibitor, lack of sufficient nuclear targeting of the inhibitor, hindrance by the YFP part, lack of access to endogenous SETD2 in chromatin complexes, other reasons).

Figure R3: Expression of YFP and YFP fused K36 super-substrate and K36M super-substrate (as putative inhibitor) in HEK293 cells. The image shows examples of the cells at day 2 after transfection showing high transfection yields and high expression levels of the YFP fusion proteins.

Figure R4: Histones were extracted from the cells shown in Figure R3 by acid extraction (left panel). Then, the global H3K36me3 was analyzed by Western blot using an H3K36me3 specific antibody (middle and right panels). No reduction of H3K36me3 was detectable after the expression of the K36M super-substrate inhibitor.

“Reviewer #2 (Remarks to the Author):

In this manuscript, Schullmacher et al. present the first substrate specificity analysis of the lysine methyltransferase SETD2. The authors successfully identify a novel “super” substrate that is methylated more robustly at the peptide level than previously reported non-histone substrates and histone substrates. Overall this is a solid body of work that warrants publishing; however, enthusiasm for this manuscript is slightly diminished by the following concerns that should be addressed prior to publication:

1. The authors report error analysis of their peptide array results (Supplemental Figure 2) and conclude that there is a “high reproducibility of the results.” However, while the authors correctly state that “most spots had an SD smaller than +/- 10%,” rough estimation from the error analysis shows that ~110/280 and ~80/280 spots had greater than 10% error. The authors should quantify from the replicates and report a bar graph with SD in the supplement. This will allow readers to understand which spots are more prone to error. As the data is currently reported, it is impossible to determine if the spots that can sometimes have a deviation of 30-40% between only two technical repeats are essential for determining the novel substrates.”

Reply: This is a very good suggestion. The requested information is now provided in Suppl. Fig. 2.

“2. The authors state on page 6, line 187, “a second substrate specificity analysis was performed (Figure 1 D-F and Suppl. Figure 2B). In this experiment, only one amino acid change led to higher methylation efficiency than the template substrate (I instead of V at position -1).” The authors should clarify this statement, as it appears that A29 has several substitutions that improve methylation, R37 has many substitutions that improve methylation, and P38 has positions that are better or the same.”

Reply: Thank you for pointing towards this ambiguity. A29 is a problem of the edge of the peptide array particularly affecting the A29 spot in one data set. This is now documented by the spot specific error levels that was introduced by your suggestion under point 1. As the A29 residue is outside of the range of specifically recognized residues, we think this point does not need further discussion in the text.

Regarding R37 and P38: As shown in Fig. 1E, R37 and P38 are among the best residues at these sites. This observations stands in sharp contrast to the results obtained with the H3 template sequence, where some amino acid exchanges led to drastic increases in the spot intensity. The text has been rewritten to better describe the data and the corresponding sentence now reads: “In this experiment, only few amino acid exchanges led to slightly increased methylation efficiencies (most prominently I instead of V at position -1)”.

“3. On page ten, lines 362, the authors state “Based on our data, FBN1 is the most efficient natural protein substrate of SETD2 that can be more efficiently methylated than H3K36, alpha-tubulin and STAT1.” This is an overstatement for several reasons. First, the only data showing methylation of FBN1 does not directly compare to histone H3. Secondly, the authors never compare to STAT1 and alpha-tubulin at the protein level. Finally, there is data that directly contradicts the notion that FBN1 is a more efficient substrate than histone H3. In Supplemental Figure 5, the ssK36-GST is methylated more robustly than FBN1, but in Figure 6, this same ssK36-GST is methylated at a level similar to the native H3 in the nucleosome context (lanes 2 vs 4). The authors need to run additional experiments directly comparing methylation of FBN1, H3, STAT1, and alpha-tubulin in order to make these claims.”

Reply: In the previous papers, it had been reported that the methylation of α -tubulin and STAT1 was weaker than that of H3. However, the reviewer is right in pointing out that FBN1 is not a better substrate than nucleosomes. This has now been clarified in the text: "...Based on our results and literature data ^{18, 19}, FBN1 is more efficiently methylated than H3, α -tubulin and STAT1, but methylation of H3K36 in the nucleosomal context is even stronger."

REVIEWERS' COMMENTS:

Reviewer #1 (Remarks to the Author):

The authors have addressed all my concerns.

Reviewer #2 (Remarks to the Author):

In this revised version of the manuscript titled "Sequence specificity analysis of the SETD2 protein lysine methyltransferase leads to the discovery of a SETD2 super-substrate", the authors provided a thorough and adequate reply to the first round of reviewer comments and included additional data and analysis to support their study. As it now stands, my enthusiasm for this study is high and I recommend publication.

**Sequence specificity analysis of the SETD2 protein lysine methyltransferase leads to the discovery of a SETD2 super-substrate
COMMSBIO-19-1945A**

Reply to the reviewers' comments:

There were no open questions left and no comments to be considered. Thank you for this positive assessment.